# Therapeutic Effect of Regional Chemotherapy in Diffuse Metastatic Cholangiocarcinoma

**DOI:** 10.3390/cancers14153701

**Published:** 2022-07-29

**Authors:** Yogesh Vashist, Kornelia Aigner, Sabine Gailhofer, Karl R. Aigner

**Affiliations:** 1Clinic for Surgical Oncology, Medias Klinikum Burghausen, Krankenhausstrasse 3a, 84489 Burghausen, Germany; vashist@medias-klinikum.de (Y.V.); s.gailhofer@medias-klinikum.de (S.G.); 2Department of Tumor Biology, Medias Klinikum Burghausen, Krankenhausstrasse 3a, 84489 Burghausen, Germany; kornelia.aigner@medias-klinikum.de

**Keywords:** cholangiocarcinoma, metastasis, regional chemotherapy, survival, hypoxic abdominal perfusion, arterial infusion

## Abstract

**Simple Summary:**

Cholangiocarcinoma are mostly diagnosed at a late stage and early recurrence is also very common with 5-year survival rates of <5% in unresectable, and 33% in resectable disease. Systemic therapy options are limited with unsatisfactory outcome. The aim of our study was to assess the efficacy of regional chemotherapy in diffuse metastatic cholangiocarcinoma. In 36 diffuse metastatic cholangiocarcinoma patients 189 cycles of regional chemotherapy using arterial infusion and perfusion techniques have been applied. Regional chemotherapy provided an excellent outcome with a median therapy specific survival of 12 months. Regional chemotherapy is effective and superior to current available and proposed therapy options in diffuse metastatic cholangiocarcinoma.

**Abstract:**

Background: Current therapeutic options in diffuse metastatic cholangiocarcinoma (CCC) are limited with unsatisfactory results. We evaluated the efficacy of regional chemotherapy (RegCTx) using arterial infusion (AI), hypoxic stop-flow abdominal perfusion (HAP), upper abdominal perfusion (UAP) and isolated-thoracic perfusion (ITP) in 36 patients with metastatic perihilar and intrahepatic CCC. Methods: Ten patients had previously undergone a liver resection and in 14 patients the previous systemic chemotherapy (sCTx) approach had failed. A total of 189 RegCTx cycles (90 AI, 74 UAP, 13 HAP and 12 ITP) were applied using cisplatin alone or with Adriamycin and Mitomycin C. A minimum of three cycles were applied in 75% of the study population. The response was evaluated using RECIST criteria with MediasStat 28.5.14. Mortality, morbidity and survival analysis were performed using a prospective follow-up database and SPSS–28.0. Results: No procedure related mortality occurred. The overall morbidity was 56% and dominated by lymph fistulas at the inguinal access site. No grade III or IV haematological complication occurred. The overall response rate was 38% partial response, 41% stable and 21% progressive disease. Median overall survival was 23 months (95%CI 16.3–29.7). The RegCTx specific survival was 12 months (95%CI 6.5–17.5) in completely therapy naive patients but also in patients who had failed a sCTx attempt previously. Conclusion: RegCTx is feasible, safe and superior to the current proposed therapeutic options in metastatic CCC. The role of RegCTx should be determined in a larger cohort of diffuse metastatic CCC patients but also at early stages especially in initially not resectable but potentially resectable patients.

## 1. Introduction

Cholangiocarcinomas (CCCs) are rare and arise from bile duct epithelium. According to their anatomical location they have different therapy approaches. Distal bile duct cancers are treated with Whipple’s procedure, whereas perihilar and intrahepatic CCC require a liver resection. Intrahepatic and perihilar CCCs are mostly diagnosed at a locally advanced, not resectable stage, or at an already metastatic stage that excludes surgery as a therapy option [1,2]. In addition, early recurrence is also common in resected patients. The overall prognosis of CCC is poor, with 5-year survival rates of <5% in unresectable, and 33% in resectable disease [3,4].

Median overall survival (OS) with the best supportive care (BSC) in unresected patients has been reported at 5 to 7 months only [5]. Although there is a lack of randomized controlled trials (RCTs) in CCC for systemic chemotherapy (sCTx), the combination of gemcitabine and cisplatin has emerged as the gold standard as the first-line sCTx in the palliative setting based on the ABC-02 (median OS 11.7 months) and BT-22 trial (median OS 11.2 months) [6,7]. Second-line sCTx standards are undefined, and the available data pinpoints towards a median OS of only six months or less [8,9]. With the rationale to deliver higher drug concentrations and less systemic side effects for unresectable and localized metastatic disease, hepatic arterial infusion/perfusion-based therapies in various forms have been described with no effect to <7 months of median hepatic arterial therapy-specific survival [10,11,12].

The regional chemotherapy (RegCTx) approach is an oncological approach with a very low toxicity profile and high tumour response due to high cytotoxic drug concentrations in an isolated perfusion bed [13,14,15,16,17,18]. In addition, the therapy can be focused on limited regions if necessary, using the same technique, e.g., hypoxic abdominal stop-flow perfusion (HAP), upper abdominal perfusion (UAP), isolated thoracic perfusion (ITP) and intra-arterial infusion. The restricted perfusion bed that is treated during RegCTx allows potentiation of drug concentration levels at the tumour site compared to sCTx, despite using up to 20–50% lower overall cytotoxic drug amount. Furthermore, the possibility to perform a chemo-filtration ensures the lowest systemic toxicity effects [12,13,14,16,18]. RegCTx efficacy has been proven in many cancers but has not been reported in advanced metastatic CCC. 

Here, we report on our institutional experience with 36 advanced metastatic CCC patients undergoing RegCTx after failure of sCTx.

## 2. Patients and Methods

### 2.1. Characterization of the Study Population

The study was approved by the Medical Ethical Committee Medias Clinic (MIRB20211001), Burghausen, Germany. In total 36 patients with metastatic CCC were enrolled in this study. Table 1 shows all characteristics of the patients. All patients presented with highly advanced metastatic CCC, out of which ten (28%) had a documented liver resection previously and presented with a diffuse relapse of the disease. Previous sCTx, primarily based on gemcitabine alone or in combination with cisplatin, had failed in 14 (39%) of the patients. One patient had undergone trans-arterial chemoembolization previously. Sixteen patients were totally therapy naïve.

Metastatic sites included, primarily, the lymph nodes including retroperitoneal and mediastinal as well as cervical stations, liver and peritoneum. One third (N = 12) of the patients either had had a choledochal stent implanted or required a stent implantation (N = 3) prior to initiation of the RegCTx at the time of admission. The Karnofsky index was below 70% and ECOG ≥ 2 in almost half of the study population.

### 2.2. Cytotoxic Drugs and Methods

#### 2.2.1. Regional Chemotherapy Techniques

Hypoxic abdominal stop-flow perfusion (HAP) is a technique that allows isolated perfusion of the abdominal region [13,17]. Figure 1a,b shows the principle of HAP and the structure of the balloon catheter used at our institute [13,14]. Drugs were chosen according to their cytotoxic potential under hypoxic conditions. Figure 1c demonstrates representatively intraoperative balloon catheter placement and blockage of the inferior vena cava and aorta for HAP. Perfusion balloon catheters are placed in the vena cava inferior and the aorta, both right beneath the diaphragm and pneumatic cuffs around the thighs, just below the inguinal region, and the peritoneal region is connected to an extracorporeal circuit. The catheters are inserted through an incision in the groin area via the femoral artery and vein. After an intra-arterial bolus infusion into the aorta, both balloons are blocked and a stopped blood flow phase with very high drug concentrations in the abdominal arterial tree is created for five minutes. Afterwards, the isolated perfusion is run for five minutes with very high drug concentration levels. Another five minutes of perfusion with inflated balloons is conducted with simultaneous chemo-filtration. After deflating the balloons, chemo-filtration is continued until five litres of substitutional volume is reached. 

Since hepatic and perihepatic tumor load was high in almost all cases, the HAP was focused to the upper abdominal region by combination with a prior upper abdominal perfusion (UAP) that was immediately followed by the above-described HAP [13,14].

For the UAP, in the first step, the venous balloon was positioned right beneath the diaphragm and the arterial balloon is placed right beneath the celiac trunk. After angiographic verification of the celiac trunk perfusion, the chemotherapeutic drugs were infused for one minute from the tip of the arterial balloon ensuring the entire cytotoxic drug was spread through the celiac perfusion bed, but mainly in the liver. Parallel to this, the venous balloon was inflated. Thereafter, the arterial balloon was immediately slipped upstream and positioned above the celiac trunk, ensuring a mesenterial stopped blood flow phase with very high drug concentrations in the upper abdomen as a consequence of this small change in arterial catheter position. The stopped blood flow phase lasted for the first five minutes.

For step two, the perfusion was run through side holes in the catheter tubes downstream of the arterial and venous balloons. This resulted in still relatively high drug concentrations in the whole abdominal perfusion bed under hypoxic condition for five minutes. The second step was identical to the HAP treatment and was also followed by chemo-filtration as described above [13,14]. UAP and HAP were performed 87 times in our patient cohort, and all were followed by chemo-filtration.

In patients who were not suitable for general anaesthesia, or a potential strong tumour necrosis post perfusion was feared with sepsis, a hepatic intra-arterial infusion (AI) was applied only, or prior to, UAP/HAP. For this, an angiographic sidewinder catheter was inserted via the femoral artery into the celiac trunk or hepatic artery. In a few cases a hepatic arterial port via the gastroduodenal artery was surgically implanted. Drugs were infused as short infusions for five to 12 min with a short-term plateau of considerably high drug concentrations in the perfusion bed. Ninety treatment cycles were applied as hepatic intra-arterial infusion (AI) combined with full anaesthesia and followed by chemo-filtration as clinically necessary.

For patients with lung, mediastinal and or cervical metastases, an isolated thoracic perfusion (ITP) was performed [15,16]. This technique was conducted with the same balloon catheters as used for UAP and HAP, but the isolated circuit was located above the balloons in the thoracic region. Pneumatic cuffs around the upper forearms reduced the perfusion bed volume and ensured high drug concentrations in the isolated perfusion circuit. Figure 2A summarizes the applied RegCTx variations and frequencies.

#### 2.2.2. Cytotoxic Drugs

For treatment under hypoxic conditions such as abdominal perfusion, Cisplatin, Adriamycin, and Mitomycin C were used as they have equal (Cisplatin) or enhanced (Adriamycin and Mitomycin C) cytotoxic potential under anaerobic conditions [19,20]. Experimental in vitro cell culture studies have demonstrated that Mitomycin C has increased cell toxicity under hypoxic conditions, and Cisplatin has equal cell toxicity under aerobic and hypoxic conditions [19,20].

Drug dosages for perfusions were 50–60 mg Cisplatin, 25–30 mg Adriamycin (cumulative maximum dose up to 600 mg), and 10–20 mg Mitomycin c (cumulative maximum dose up to 60 mg), respectively. Intra-arterial infusions were conducted with 30–40 mg Cisplatin alone or with 10–30 mg Adriamycin, and 10–20 mg Mitomycin. Drugs were allotted at higher levels to infusions that had been followed by chemo-filtration compared to infusions without chemo-filtrations.

#### 2.2.3. Treatment Cycles

Regional chemotherapy was applied in treatment cycles. Each treatment cycle consisted of either one isolated perfusion, or one intra-arterial treatment followed by chemo-filtration or intra-arterial infusion with total cycle dosage distributed to four sequential days without chemo-filtration. Each therapy cycle was followed by a three-week therapy-free interval.

In total, 189 cycles were applied consisting of 12 ITP, 13 HAP, 74 UAP and 90 AIs. In 52% of the patients, only a single RegCTx technique was applied, whereas in the remaining patients mostly a combination of two, and in very few cases, three techniques were applied over the course of the entire treatment period. Table 2 shows the distribution of applied cycles. A minimum of three cycles were applied in 75% of the patients. The techniques were alternated for different cycles for each patient if different metastatic locations were to be treated.

For statistical analysis, SPSS for Windows (Version 28.0) was used. RegCTx-specific survival and OS curves of the patients were plotted using the Kaplan–Meier method and analysed using the log-rank test. Significant statements refer to *p*-values of two-tailed tests at *p* < 0.05. Results are presented as median survival in months with 95% confidence interval (CI). The OS was computed as the time period from the date of first diagnosis to the date of death or last follow-up, whichever occurred first. RegCTx-specific survival was defined as the time period from the date of first RegCTx to last follow-up or date of death, whichever occurred first. Patients alive at the last follow-up date were censored.

The response evaluation under RECIST criteria was undertaken with the MediasStat software version 28.5.14 and addressed as partial response (PR), stable disease (SD) or progressive disease (PD). In addition, quality of life (QoL) was assessed based on an institution specific questionnaire including nausea, vomiting, hair loss, diarrhoea, mucosal changes, fatigue, exhaustion and loss of appetite [21]. QoL was assessed during the in-hospital stay pre and post RegCTx treatment.

## 3. Results

No RegCTx-associated mortality occurred. One patient with PD developed a septic condition after the third AI cycle on day eight, and died. The overall morbidity rate was 56%, being dominated by the development of lymph fistulas at the inguinal access site; N = 14 (39%) patients, over the entire treatment duration. However, all fistulas were treated successfully conservatively. Due to the development of wound haematomas, three patients required an operative wound revision. Incidence of general side effects such as nausea and fatigue were very low, and only mild, and did not require any additional medication to post RegCTx standard protocol. Four (11%) and three (8%) patients presented a temporary grade II leuko- and thrombocytopenia, respectively, during the entire treatment course. Hair loss, hand-foot syndrome and neuropathy did not occur. No grade three or four haematological complications occurred.

Responses to the treatment were measured under RECIST criteria. Usually, after two cycles of therapy, a CT scan was conducted. In total, 61 scans were available for response evaluation. The overall response rate was 38% PR, 41% SD and 21% PD for the RegCTx in the entire cohort and over the entire period of follow-up. Cycle-specific response demonstrated the best PR and SD after the third cycle, both declining constantly and substantially after the fifth cycle. On the other hand, PD was less frequent in the earlier cycles but picked up after the third cycle. Figure 2B depicts the response evaluation.

The institute-specific QoL indicator as a clinical response parameter was documented during the in-hospital stay pre and post RegCTx treatment. The overall clinical response (QoL) evaluation yielded a CR in 8%, PR in 31%, SD in 43% and a PD in 18% of the patients.

The median OS was 23 months (95%CI 16.3–29.7) with 14 (39%) patients having a survival >2 years. Survival rates reached 67, 42 and 28% at years 1, 2 and 3 respectively.

In addition, the effect of RegCTx resulted in a median survival benefit, from the time of initiation of the RegCTx, of 12 months (95%CI 6.5–17.5) and 10 (28%) patients were alive even 15 months under the RegCTx. The according survival rates were 44, 17 and 11% at years 1, 2 and 3 respectively. Figure 3 and Figure 4 show the OS and RegCTx-specific survival in all 36 patients.

Sub-analysis determined whether a liver resection or a sCTx was previously performed or not, and demonstrated a median survival benefit of 20 months (95%CI 6.2–33.7) and 22 months (95%CI 6–38), respectively, in patients without previous liver resection or sCTx compared to 26 months (95%CI 7.4–45; *p* = 0.131) and 26 months (95%CI 0.9–51.1; *p* = 0.621) with liver or previous sCTx ,respectively.

To further determine the impact of RegCTx alone in metastatic CCC, we performed a survival sub-analysis in sCTx-naïve patients without any liver resection. This also resulted in a substantial median survival of 12 months (95%CI 0–29) in 16 (44%) of the patients.

## 4. Discussion

The majority of CCC cases are at the time of first presentation either locally advanced and not resectable or already metastatic [22,23]. Surgical resection remains the curative cornerstone; however, only a fraction of patients are eligible for surgery, and an R0-resection is again achieved in a fraction of those patients only, with a recurrence rate amounting >60% at a median follow-up of 12 months [3,24]. Introduction of neoadjuvant therapy and advanced complex surgical techniques, e.g., portal vein embolization, extended resections and procedures such as ALPPS [25,26], have certainly expanded the therapeutical spectrum with a survival benefit for a fraction of CCC patients; however, at recurrence or in a metastatic condition, the therapeutic options are still very limited, especially as established first-line treatment has been missing recently due to lack of RCTs in CCC. Over the past 15 years, many trials, e.g., BILCAP, PRODIGE-12, ABC-02 and BT-22, have been carried out to clarify the role of adjuvant, first-line, second-line and, very recently, even targeted therapies in CCC [6,7,24,27].

However, the overall outcome remains still unsatisfactory with survival rates between five to 11 months only in recurrent or primary unresectable and metastatic disease [3,28]. In addition, severe systemic toxicity and surgery-related morbidity and mortality remain unsolved issues in the case of extended resections, cholestatic patients and limited future liver remnants, as well as impaired liver function. Hence, most treatment options are only applicable to patients with overall good general condition (ECOG 0–1) and limited tumour load [25,26].

We report in a cohort of 36 patients with advanced metastatic perihilar or intrahepatic CCC patients, of whom 16 (42%) had previously already undergone a sCTx and 10 (26%) liver resectional procedures, that RegCTx is a safe and viable approach with a superior outcome of RegCTx-specific survival of 12 months compared to the current available therapeutic approaches [3,23].

Historically the natural course of CCC under BSC has been reported to be around five months only in treatment of naïve patients, with a performance status of ECOG 0–2 [1,3,23]. We were able to demonstrate that RegCTx is applicable even in patients with ECOG 3 with the lowest toxicity profile, and a median OS of 12 months in complete therapy-naïve diffuse metastatic CCC patients.

Only a decade ago, no survival benefit was suggested for chemotherapy versus BSC in biliary tract cancer [3]. Based on a pooled analysis of 161 trials, the best outcome had been reported with a gemcitabine and platin combination [29]. The ABC-02 trial, along with the BT-22 study and several other studies, demonstrated the superiority of gemcitabine and cisplatin combination with median OS of a little more than 11 months, and progression-free survival between 6–8 months [6,7]. In none of the studies, a BSC arm was present, but the combination compared to gemcitabine alone. Although several other studies have been carried out, the outcome with the gemcitabine and cisplatin combination remained below 12 months in all reported trials [3]. In almost 40% of our patients, either gemcitabine alone or gemcitabine and cisplatin-based sCTx had been previously applied.

In our cohort with almost half of the patients having an ECOG score of ≥2, the median OS was 23 months and RegCTx-specific survival was 12 months. It needs to be pointed out that in our cohort only perihilar and intrahepatic CCCs were included, whereas in the ABC-02 trial >40% of the study population had distal bile duct, ampullary or gall bladder cancer [7]. The reported response rate of 81% (PR and SD) in the ABC-02 trial is comparable to our 38% PR and 41% SD. In line with this, the RegCTx-specific survival amounted to 12 months in median, and in our patient cohort chemotherapy-related toxicity side effects were seldom, and appeared only in a few patients as grade II, whereas grade III and IV haematological toxic side effects in the ABC-02 trial were reported in one third of the study population [7]. In addition, 14 patients in our cohort had previously either undergone a gemcitabine, or a gemcitabine plus cisplatin, sCTx. In this subset, the RegCTx-specific survival was in median 12 months, indicating a clear RegCTx-directed survival benefit in those patients who primarily had failed in the sCTx approach. The BT-22 study had reported similar results to the ABC-02 trial, with a heterogeneous study population and grade III and IV toxic side effects in up to almost 40% of patients [6]. The reason for the low toxicity profile with our RegCTx approach is based upon the frequent combination of RegCTx with chemo-filtration on the one hand, but also using less total dosages and directed therapy towards a limited perfusion bed to reduce collateral damage to other organs, hence reducing cumulative toxicity over time and ensuring a longer treatment phase if required.

The best survival was demonstrated in patients undergoing surgical resection, with curative intent, with survival rates up to 37 months. Unfortunately, only <20% of CCCs are eligible for surgery [22,23,28]. Accordingly, the outcome in resected patients receiving the same therapy as unresected patients results in a much better survival for resected patients, as demonstrated by the BILCAP and PRODIGE-12 trial [24,27].

In our study population, ten patients had undergone liver resection, and the survival of those patients is in line with the reported effect of surgery followed by sCTx. Patients having had a liver resection prior to RegCTx had a median OS of 26 months compared to 20 months only in those who did not undergo any surgical procedure prior to RegCTx.

Options following failure to a first-line systemic approach are highly limited and practically not evident in CCC [3,23]. Second-line treatment studies have demonstrated median OS of around six months only. The most commonly reported regimes were FOLFIRI, FOLFOX, XELOX, capecitabine and XELIRI [8,9,30]. Since in our cohort 14 patients had undergone sCTx previously, we were able to demonstrate that RegCTx can be safely applied in those patients in whom first-line treatment had failed. In that subset of patients receiving RegCTx as a second line treatment, the RegCTx-specific survival was 12 months in median, and the OS was 26 months, showing RegCTx being superior compared to all other currently available options.

RegCTx in the form of hepatic arterial infusion and perfusion has been reported by other groups with smaller study population sizes, data merging from multiple institutions, higher toxicity profiles, and poorer outcome. Kasai et al. reported a median OS of 14 months in a cohort of 80% non-metastatic patients [12]. Marquardt et al. reported on 15 patients from nine different institutions with 50% grade 3–5 complications and a median OS of 7.6 months only [10]. In contrast, our patients were treated at a single centre, with a standardized technique, with low morbidity, especially a low haematologic toxicity profile, and superior outcome.

Along with the survival benefit in these highly advanced metastatic patients, the benefit in terms of QoL with RegCTx lie in the shorter treatment period and low toxicity profile. Patients undergoing RegCTx required a hospital stay of only a few days per cycle, and 75% of our patients received at least three cycles. Our data demonstrate that the first three cycles were the most efficient, with the best documented disease response. In our cohort, patients were able to move around on the day of RegCTx and no documented side effects appeared that required re-hospitalization. Patients could be discharged after each cycle within three to five days from the hospital, followed by a three-week therapy-free interval. Since QoL is a major aspect in the management of metastatic patients, our approach ensured no impairment in QoL but a significant improvement. Taken into consideration that we treated ECOG 2 and 3 patients, this points towards justification of this approach without even taking the OS benefit and low toxicity profile into account.

Limitations of the current study are the long inclusion period of two decades, numbers of therapy cycles applied, the heterogenous group with regard to pre-treatment, and non-standardized QoL assessment.

Besides the RECIST response evaluation, we also considered QoL estimation based on a subjective questionnaire (patients’ own remarks) [21]. This is clearly not objective or reported in a standardised manner and should be assessed with a formal QoL questionnaire in the future. 

The role of sCTx in CCC has only emerged slowly over the past two decades, hence few patients might have had a better survival if sCTx would have been available earlier. However, we were able to demonstrate that therapy-naïve patients had a clear indisputable benefit with RegCTx. In addition, we only included perihilar and intrahepatic CCC patients in our study cohort and intentionally excluded gall bladder and distal bile duct and ampullary cancer; hence, we present a clinically homogeneous metastatic cohort. In line with this is the pre-treatment of the included patients. Taken the incidence of the disease and the limited therapy options, we present the largest series of metastatic CCC patients treated with RegCTx. This is a highly individualised concept; hence, adaptation with regard to form and frequency is essential and also dependent on the patient’s own will to continue or terminate a specific therapy.

## 5. Conclusions

In conclusion, RegCTx offers a low toxicity-associated therapy approach in highly advanced metastatic CCC patients with a clear survival outcome that is superior to other currently available therapy options. Future studies are required to evaluate the role of RegCTx in the multimodal management of CCC and in less advanced disease stages, especially in primarily non-resectable but potentially resectable cases.

## Figures and Tables

**Figure 1 cancers-14-03701-f001:**
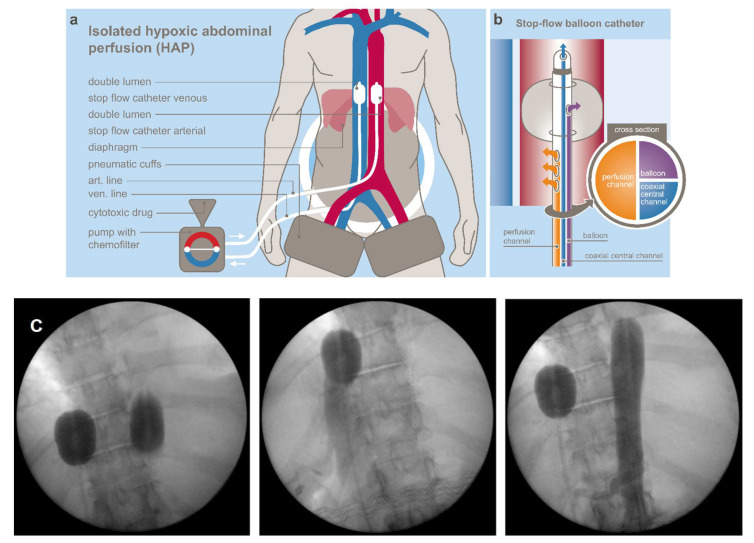
(**a**) schematic principle of isolated hypoxic abdominal perfusion. (**b**) Structure of stop-flow balloon catheter. (**c**) Intraoperative images of balloon catheter placement, inferior vena cava and aorta blockage for HAP.

**Figure 2 cancers-14-03701-f002:**
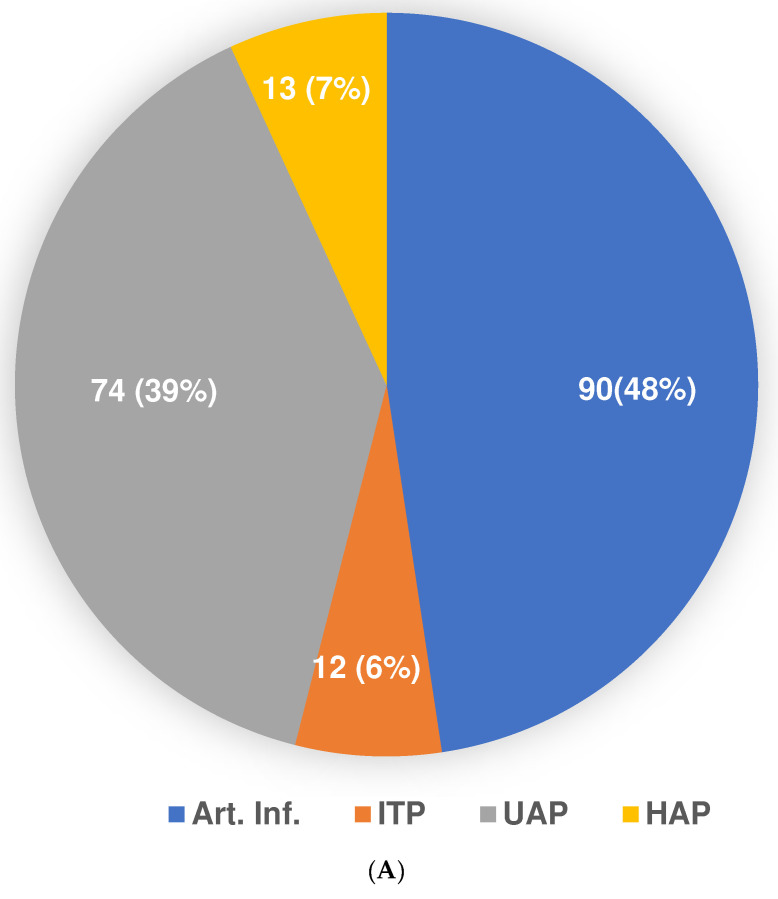
(**A**) Distribution of type of regional chemotherapy approach in the applied 189 cycles. (**B**) Response evaluation according to RECIST criteria based on 61 CT-scans.

**Figure 3 cancers-14-03701-f003:**
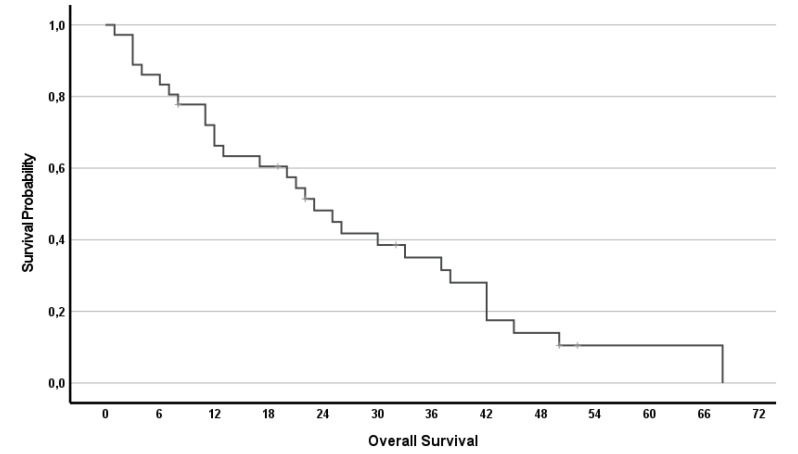
Overall survival of all 36 patients.

**Figure 4 cancers-14-03701-f004:**
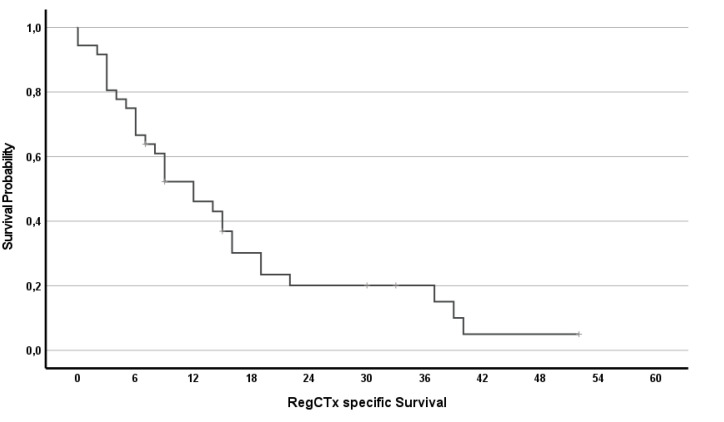
Regional chemotherapy-specific survival of all 36 patients.

**Table 1 cancers-14-03701-t001:** Patients’ characteristics.

Variable	N	%
All	36	100
Sex		
female	15	41.7
male	21	58.3
Metastatic site		
liver	22	61.1
lymph nodes	36	100.0
lungs	8	22.2
peritoneum	21	58.3
others	5	13.9
Karnofsky index		
100–70	19	52.8
60–50	10	27.8
40–30	7	19.4
ECOG		
0–1	19	52.8
2	10	27.8
3	7	19.4
Liver resection		
	10	27.8
SCTx		
	14	38.9
Choledochal Stent		
	12	33.3
RegCTx		
Total Cycles	189	100.0
Art. Infusion	90	47.6
UAP	74	39.2
ITP	12	6.3
HAP	13	6.9

**Table 2 cancers-14-03701-t002:** Applied Cycles.

Number	N	%	Cumulative %
1	2	5.6	5.6
2	7	19.4	25.0
3	3	8.3	33.3
4	7	19.4	52.8
5	4	11.1	63.9
6	4	11.1	75.0
7	1	2.8	77.8
8	2	5.6	83.3
9	2	5.6	88.9
11	1	2.8	91.7
12	1	2.8	94.4
14	2	5.6	100.0

## Data Availability

The data presented in this study are available in this article.

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
