# Peer review of "Therapeutic Effect of Regional Chemotherapy in Diffuse Metastatic Cholangiocarcinoma"

_cancers, 2022, doi:10.3390/cancers14153701_

Round 1

Reviewer 1 Report

This is a very well written and comprehensive article on the subject of regional chemotherapy in patients with metastatic cholangiocarcinoma. Authors will agree with me that these procedures will only be performed in highly specialised and selected centres. can the authors make an estimate on how many patients might benefit from such treatment? and what further studies are required for this to be adopted in the wider clinical practice? 

Author Response

Reply: Thank you for those positive remarks. The procedure requires certainly some skills but does not represent anything that cannot easily be established in a successful HPB unit. Comparing to a complex liver surgery the surgical act for regional chemotherapy is of minor importance however vascular anatomy and even more selection of chemotherapy and drug dosages as well as the postoperative management but utmost performing regional chemotherapy in the right patient are the true challenges to master hence it requires training at a centre certainly.

With the regard to the number of patients we need to reflect the aggressive natural course of CCC despite R0 resection. Unfortunately, only a fraction of patients are resectable at the time of diagnosis and majority suffers of early relapse despite resection hence this technique could be applicable in majority of CCC patients – those who may be subjected to regional chemotherapy for downsizing and reach resectability and also those who already suffer of diffuse spread. Accordingly clinical evaluation in larger cohorts of potential resectable and metastatic CCC is warranted.

Reviewer 2 Report

The manuscript evaluated the efficacy of regional chemotherapy (RegCTx) using arterial infusion (AI), hypoxic stop-flow abdominal perfusion (HAP), upper abdominal perfusion (UAP) and isolated-thoracic perfusion (ITP) in 36 patients with metastatic perihilar and intrahepatic cholangiocarcinoma (CCC). The Authors found that RegCTx is feasible, safe and superior to the current proposed therapeutic options. The work is very interesting but some points need to be improved:

-     - In the Introduction section, they described the different therapy approaches for cholangiocarcinoma in relation to the anatomical location. Could they explain better the different anatomical location, histological aspects and origins of CCC?

-      - In the figure 2, please add the letters A and B in the two different pictures described in the legend.

-     - In the first part of the Results section, they explained the morbidity rate in several patients. Could it be related to the different origin of CCC or to other specific features of the patient? Please speculate about that.

Author Response

Reviewer 2:

The manuscript evaluated the efficacy of regional chemotherapy (RegCTx) using arterial infusion (AI), hypoxic stop-flow abdominal perfusion (HAP), upper abdominal perfusion (UAP) and isolated-thoracic perfusion (ITP) in 36 patients with metastatic perihilar and intrahepatic cholangiocarcinoma (CCC). The Authors found that RegCTx is feasible, safe and superior to the current proposed therapeutic options. The work is very interesting but some points need to be improved:

Reply: Thank you for the positive evaluation.

-     - In the Introduction section, they described the different therapy approaches for cholangiocarcinoma in relation to the anatomical location. Could they explain better the different anatomical location, histological aspects and origins of CCC?

Reply: We did describe the clinically relevant classification of different types of bile duct cancer. Our description intended to point out that we have not included distal bile duct cancer or gall bladder cancer with different outcome but only perihilar and intrahepatic cancer in this study. We do apologize but we do not understand the remark of the reviewer regarding elaborating more on location and histological type.

-      - In the figure 2, please add the letters A and B in the two different pictures described in the legend.

Reply: We have done it.

-     - In the first part of the Results section, they explained the morbidity rate in several patients. Could it be related to the different origin of CCC or to other specific features of the patient? Please speculate about that.

Reply: We do not believe that the morbidity was related to type of cancer. Lymph fistulas are common after vascular dissection in the inguinal area and have been conservatively treated. The wound revisions were due to wound haematomas which result from full systematic heparinization that is necessary for perfusion procedures.